# The Influence of Hard Coatings on Fatigue Properties of Pure Titanium by a Novel Testing Method

**DOI:** 10.3390/ma17040835

**Published:** 2024-02-09

**Authors:** Cai Hu, Lei Zhao, Yong Zhang, Zhinan Du, Yunlai Deng

**Affiliations:** 1School of Materials Science and Engineering, Central South University, Changsha 410083, China; 22022055@csu.edu.cn (C.H.);; 2Manufacturing Technology Institute, Aviation Industry Corporation of China, Ltd., Beijing 100024, China; 3Beijing Satellite Manufacturing Co., Ltd., Beijing 100094, China

**Keywords:** hard coatings, fatigue properties, pure titanium, fatigue cracks

## Abstract

This study investigates the impact of hard coatings on the fatigue properties of pure titanium. A specialized fatigue test which ensured machine equivalence was conducted to compare the fatigue behavior of coated and uncoated metals. The findings reveal that the application of coatings adversely affects the fatigue properties of pure titanium due to stress concentration from the coating, which accelerates fatigue crack propagation within the substrate material. Notably, zigzag fatigue cracks at the interface between the coating and substrate and multiple micro-cracks initiated within the coating are found.

## 1. Introduction

Titanium has extensive applications across multiple sectors, including aerospace, nuclear power generation, marine science, the automotive industry, and medical fields. Its exceptional attributes, such as high specific strength, low density, excellent mechanical properties, toughness, corrosion resistance, and ability to withstand high temperatures, contribute to its widespread use [1].

Fatigue properties play a crucial role in the reliability and performance of metallic materials, particularly in applications where cyclic loading is involved. To enhance the fatigue resistance of metals, the application of hard coatings has gained significant attention in recent years. Hard coatings have been extensively used to improve the surface properties of metals, such as wear resistance, hardness, and corrosion resistance. However, the impact of hard coatings on the fatigue behavior of metals, particularly pure titanium, remains a topic of interest and investigation.

Despite its numerous advantages, titanium is vulnerable to adhesion wear when in contact with grinding materials due to its limited thermal conductivity. This may lead to the wear failure of components [1]. To overcome these challenges, the implementation of hard coatings is employed. These coatings serve to enhance surface hardness and prevent the substrate from being lost during friction and wear. Such hard coatings can also be found for aluminum, steel, and many other materials [2,3,4], protecting the substrate metals from damages [5,6].

Although coatings have numerous advantages, there is ongoing debate among scholars regarding the influence of hard coatings on fatigue properties.

Several researchers have asserted that coatings can enhance the fatigue properties of metals [7,8,9]. For instance, Liu observed that coatings exhibit a suppressive effect on the fatigue crack initiation process, attributed to the distinctive plastic deformation behavior of thin films [10]. Brame’s findings indicated that hard coatings can impede the accumulation of dislocations on the substrate’s surface by virtue of their high strain energy, thereby inhibiting plastic deformation of the substrate [11]. Kovacı argued that hard coatings improve fatigue crack life by augmenting fatigue crack growth resistance through the generation of surface residual stresses and increased surface hardness [12]. These researchers view coatings as a means of reinforcing metals, attributed to the coatings’ high mechanical strength and substantial compressive residual stress. When there is relatively good adhesion between the coating and the substrate, these factors synergistically contribute to the delay of crack propagation towards the substrate. As a result, hard coatings effectively enhance the fatigue life of substrate metals. Compared to uncoated 316 L stainless steel, the fatigue life and fatigue strength of TiN coatings can be increased up to 1677% and 10.8% using the Basquin equation [13]. Even a 20% increased rate can be found in the fatigue strength of the stainless steel JIS SUS316 if DLC (diamond-like carbon) coating. This number can be 53% when plasma-carburized treatment is added [8]. SEM fractographs show that the fracture surface is flat, which indicates that fracturing of the substrate occurs as a consequence of propagation of a single crack [7,13]. These demonstrate that the substrate under a hard coating could not follow the plastic deformation at the crack tip. The effect of the microstructure of the substrate has not been studied clearly. But the enhancement of hard coatings depends on the substrate metals. This effect on steel is greater than that on Ni-based alloys [10]. Nevertheless, if the adhesion of the hard coating is poor, it easily causes coating delamination or cracking, which lead to the fast fatigue failure of the hard coating [14].

However, other researchers have reported that coatings can easily fracture under stress and cause micro-cracking of the underlying metal substrate, which may reduce fatigue properties [15,16,17,18,19,20,21,22]. Bai’s study found that coating fractures induced cleavage cracking of the substrate, which was responsible for the fatigue crack initiation, as the applied stress increased above the critical stress [16]. Zhang indicated that the reduced fatigue performance could be attributed to the cracking of coatings, which accelerated fatigue failure by inducing stress concentration and reducing crack initiation cycles [23]. During the fatigue process, the initiation of the crack can occur at different areas. Fatigue cracks are initiated at the interface between the coating and the substrate, as they have been found in WC-10Co4Cr coatings on AISI 4135 steel substrates [19] and at the surfaces of TiN coatings on TC11 substrates [22]. Internal metallic inclusions or defects can be sources of fatigue cracking [22]. Based on the “notch effect” model [2,24,25] or the “film-induced cracking of substrate” model [26], the coated metals exhibit low fatigue properties. The percentage of reduction in fatigue properties is related to the coating’s pores and micro-cracks. But this negative effect on fatigue properties can be reduced via the integration of ultrafine powders or reducing the spraying distance, as well as during the process of coating preparation [20]. A coating with a dense morphology and relatively lower porosity is preferred. The thickness of a coating can also have an influence on fatigue properties [27]. The thicker the coating, the stronger the stress concentration, which can lower the fatigue properties [16] as well as the tensile properties [19]. 

The impact of hard coatings on the fatigue properties of metals has been extensively studied, resulting in a wealth of comprehensive data. However, two contradictory opinions are popular in academic papers, and the mechanism of hard coatings’ influence remains unclear. Ensuring machine equivalence in fatigue tests poses a challenge, as even slight stress fluctuations can significantly affect the measured fatigue life. This challenge stems from factors such as assembly errors, clamping force, test order, and test duration. Given the inherent randomness of fatigue test results, a significant number of specimens must be prepared to obtain reliable and convincing outcomes. Typically, more than a dozen specimens are required to determine the fatigue limit, while constructing the S-N curve necessitates twenty specimens. These numbers double when comparing the fatigue properties between coated and uncoated specimens [28].

In order to overcome the challenges in comparing the fatigue properties of coated and uncoated specimens, this study proposes a novel type of fatigue specimen which eliminates the fluctuations from the testing devise. This approach enables a more effective comparison of fatigue properties between coated and uncoated specimens and requires fewer specimens, making it more economical and easier to prepare. Apart from these advantages, various types of materials for the substrate and hard coating can be applied to this structure in further studies.

In this paper, tungsten carbide–cobalt (WC-Co) coatings are employed due to their extremely high hardness, tensile strength, elastic modulus, and chemical stability, which are crucial to resisting external friction, wear, and corrosion. The fatigue properties of pure titanium with and without WC-Co coatings are compared using the newly designed specimens. To elucidate the mechanism underlying the impact of hard coatings on the fatigue limits of pure titanium metals, a comprehensive microstructure analysis is conducted. The detailed characterization provides valuable insights into the relationship between hard coatings and fatigue performance.

## 2. Experimental Procedures

### 2.1. Specimen Preparation

The proposed fatigue specimen is ring-shaped and comprises a pure titanium substrate with the same thickness as the WC-Co coating applied to one half of the ring. It is presented in Figure 1a. Prior to thermal spray coating, grit blasting with aluminum oxide is performed to enhance adhesion. The ring has a radius of 10 mm, a wall thickness of 2 mm, and a length of 8 mm. This specimen can be readily produced by using a substrate metal tube with a coating that is subsequently cut into multiple pieces. The dimensions of each specimen can be adjusted according to equipment or preference. A cyclic load is applied to the diameter direction of the ring-shaped specimen, which is pressed by the chuck of the fatigue machine, with a plastic tie looped through the ring for safety. The two sides of the specimen can be viewed as two parts: the substrate and the substrate with the coating. As pressure is applied, both sides of the specimen experience equal bending moments due to this structure. Zones A and B are located at the outermost regions of the two curved beams. One of these zones will fracture first, revealing a comparison of the fatigue resistance of metal with and without coating. To compare the changes in the microstructure of the specimen before and after the fatigue test, one side of the cutting surface was polished using sandpaper and colloidal silica. Scanning electron microscope (SEM) checks before the test showed no evident existing cracks on any part of the specimen. As shown in Figure 1b, the thickness of the WC-Co coating was about 311.5 μm, and the coating was nearly completely dense. As shown in Figure 1c, the specimen was placed between the two chucks of the fatigue machine, which was only subjected to pressure load. For safety reasons, a plastic strap ran through it to prevent the specimen from flying out during the test.

The distribution of Mises stress in the ring-shaped specimen under fatigue load was analyzed through FEM simulation by Abaqus 2022. A linear elastic constitutive model with material constants, listed in Table 1, was used for the specimen in this study. A small elastic deformation of the specimen was achieved by displacing two rigid plates in the loading direction. The model consisted of a total of 6756 elements and 41,552 nodes, representing a 3D geometry. No symmetry was utilized during the modeling process. The finite element analysis employed the C3D20R element type. Gaussian integration was used for numerical integration calculations, employing 20 nodes and 27 integration points. The critical region of the model was the contact interface between the ring and the coating. To ensure perfect contact between the ring and the coating, a contact surface was defined and appropriate boundary conditions were applied. A finite element mesh with approximate global size of 6 was applied, which is shown in Figure 2a, demonstrating a high-quality structured mesh that minimized stress singularities and facilitated subsequent FEM computations. Considering the convergence of the numerical model, the use of quadratic elements allow better representation of a curved boundary and have more compatibility in calculation, compared to linear elements. These provided more accurate strain and stress values while maintaining reliable displacement results. Additionally, given the presence of curved surfaces within the model, particularly at the critical contact interface, the use of quadratic elements enabled better approximation of curved geometries. Moreover, quadratic elements exhibited faster convergence rates compared to linear elements, converging more rapidly towards the exact solution when the mesh was refined. The choice of using full integration instead of reduced integration ensured that the finite element solution converged monotonically to the exact solution as the element size decreased. By utilizing C3D20R elements, the occurrence of shear locking issues was effectively minimized, thereby ensuring accurate stress calculations.

The Mises stress distribution is depicted in Figure 2b. The highest stress occurred in the coating within zone B. In comparison to zone A, the stress in the substrate beneath the coating was significantly reduced in zone B, thanks to the protective effect of the coating. Figure 2c,d illustrate the through-thickness normal stress along the loading direction in zones A and B, including both tensile stress on the outside and compressive stress on the inside. The compressive stress was nearly identical in both zones. However, there was a sudden increase in tensile stress from the substrate to the coating in zone B. The maximum tensile stress in zone A was approximately 250 MPa, while it exceeded 700 MPa in zone B. Nevertheless, the simulation results regarding stress distribution could not determine which zone has a longer fatigue life. The experimental results will provide the answer to this question.

### 2.2. Experimental Procedures

In this study, fatigue testing of the ring-shaped specimens was conducted on a SDS100 Electro-hydraulic servo fatigue testing machine (Guanteng Automation Technology Co., Ltd., Jilin, China) at room temperature using a sinusoidal load with a frequency of 20 Hz, ranging from −0.5 kN to 0 kN. The test was terminated upon the occurrence of high cyclic fatigue failure in either zone A or zone B. This approach allowed us to compare the fatigue life between these two zones. By comparing the fatigue life in different zones, we aimed to gain insights into the underlying mechanisms that govern the fatigue behavior of the specimens. To further investigate the fracture mechanism, the microstructure of the fractured specimens and polished cutting surfaces were carefully examined using a ZEISS M10A scanning electron microscope (ZEISS Group, Jena, Germany). By analyzing the microstructure, any microstructural changes, cracks, or other indicators that could provide insights into the fatigue failure mechanisms were identified.

## 3. Results and Discussion

### 3.1. Results of Fatigue Test

After 3.6 × 10^6^ cycles, a crack was observed in zone B. Figure 3a shows a specimen after the fatigue test; the fracture occurred at zone B.

Another fracture zone was observed at the bottom of the specimen on the loading axis. SEM showed that no cracks were found in zone A, as shown in the image in Figure 3b. This phenomenon revealed that the pure titanium substrate with WC-Co coating had less fatigue resistance than pure titanium. The WC-Co hard coating reduced the fatigue properties of pure titanium, and it can be concluded that this WC-Co hard coating on pure titanium had a negative effect on fatigue properties under these conditions. 

### 3.2. Fractography Characterization

Fractography characterization was performed on zone B, and an overview of the fracture surface is presented in Figure 4.

The surface exhibited a typical fibrous pattern associated with fatigue fracture, with fatigue striations present, representing the stepwise progression of crack propagation known as stage II fatigue. The crack was initiated at the interface between the coating and substrate, and then propagated through the cross-section of the specimen. The fracture was primarily caused by a single crack. The fatigue striations are clearly visible on the titanium substrate, indicating the progressive crack growth. However, no striations were observed on the WC-Co coating. The coating exhibited a brittle fracture mode and served as the initiation site for the fatigue crack. Despite its high hardness, the coating was relatively thin compared to the substrate, but still underwent deformation in this region due to its integrity. WC-Co had a higher Young’s modulus than pure titanium, resulting in the coating experiencing the maximum local stress on the specimen. However, it possessed minimal plasticity, leading to fracture occurrence in the coating despite its hardness.

Figure 5 shows the interface of the Ti substrate and the WC-Co coating. The striations in Figure 5a revealed the propagating path of the fatigue fracture. The origin of this crack was at the interface of coating and substrate, located in a dark area. The formation of this area was caused by the fractured coating being higher than the substrate. In other words, the crack crossed from the coating to the substrate in a zigzag path. A higher substrate and lower coating are shown in Figure 5b. Fatigue bands can be seen clearly in the Ti substrate. C and D planes can be found according to the band pattern. These two planes were not parallel and exhibited an angle to each other. A zigzag fatigue propagation occurred in this area.

This phenomenon is a result of adhesion between the coating and substrate. The crack followed the path with the weakest adhesion area. The tough surface after grit blasting increased the contact area between the coating and the substrate, which exhibited strong bonding in ups and downs at the interface, as seen in Figure 1b. However, this embedded structure made the soft substrate more susceptible to fatigue conditions, especially after hard coating fracturing. Figure 6 illustrates this mechanism in steps.

The initial condition of the interface with ups and downs is shown in Figure 6a. A cyclic load F was applied to the coating and substrate. Due to the ups of the substrate, the thickness of the coating here was reduced, and fatigue cracks became able to easily cross the coating through this part, as shown in Figure 6b. After fracturing of the coating, the load F on the coating was released, and the reaction of this system changed. Based on the micro-movement caused by the load F, three areas—α, β, and γ—could be divided, as shown in Figure 6c, assuming that the right side of the specimen was fixed.

The left part of the coating is α, called the large-motion area. It had reciprocating motion just following the fatigue machine due to its high Young’s modulus and the crack. However, the substrate had homogeneous elastic deformation from right to left. Compared to the α area, γ had less reciprocating motion, and was called the medium-motion area. The β area was connected to the fixed edge at the right, which had the least motion. Compared to the γ area, β was called the low-motion area. The coating and substrate shared movement from the fatigue machine differently. The motion of the coating could only be seen as the width of the crack changing, while the substrate had uniform deformation. Consequently, a stress concentration area in the substrate at the tip of the coating crack was formed. Meanwhile, the downs of the substrate were fulfilled by the coating, which moved just like the α area. There was a difference in the motion of the coating and the substrate. At the root of the downs, the substrate was vulnerable because the coating was much harder than the substrate. A void was formed at the substrate due to larger reciprocating motion of the coating.

Until this step, weak points appeared at the ups and downs. As shown in Figure 6d, it is reasonable that fatigue cracks propagate by connecting these two points first and then crossing deeply into the substrate. Eventually, a zigzag path with fatigue bands was formed in this plastic substrate material. Additionally, this theory explains why zigzag fatigue cracks and more vulnerable substrates coexist.

After the fatigue test, optical microscopy (OM) was performed on the polished surface. The examination found a zigzag crack, as shown in Figure 7, which further substantiates the theory presented.

The vulnerable areas were observed to be extensively distributed along the interface of the specimen, primarily due to the irregularities and undulations present on the rough substrate surface. Specifically, the cracks within the coating were found to be connected to the ups, and the cracks within the substrate to the downs. This observation highlights the critical role played by the surface topography in influencing crack propagation and distribution within the coating–substrate system.

### 3.3. Microcracks

The specimen had an evident crack at zone A. Besides the existing fractured area, several microcracks could be found at zone B. Figure 8 shows the cross section of the WC-Co coating.

This illustrates that the fatigue crack was initiated at the surface of the coating, which was also found in Zhang’s study [22]. The propagation of the fatigue crack penetrated from the surface of the coating towards the substrate, leading to continued cracking in the substrate. This crack did not propagate throughout the thickness of the specimen, but it stopped due to stress releasing when another crack successfully fractured, which generates the fracture surface shown in Figure 3. Cracks in hard coatings have a negative effect on fatigue properties by accelerating the process of crack initiation. The premature cracking of the substrate was due to the high stress concentration at the tip of the coating crack. A similar phenomenon was found with film-induced cracking of the substrate [15]. The nucleation of fatigue cracks in the substrate under stress concentration can be explained by the formation of persistent slip bands (PSB) [26]. These slip bands are localized areas where plastic deformation occurs, and they act as preferential sites for crack initiation. Stress concentration at the tip of the slip bands can lead to the nucleation of fatigue cracks in the substrate material.

Bai [16] proposed a coating-cracking-induced low cyclic stress substrate damage model (CCILCSSD) to explain this mechanism. The stress concentration in the substrate material under a fractured coating can be identified by the stress concentration factor $\alpha_{\sigma }$:(1)$$\alpha_{\sigma }=\frac{\sigma_{max}}{\sigma_{n}}=1+2\sqrt{\frac{h}{\rho }}$$
where $\alpha_{\sigma }$ is the stress concentration factor, $\sigma_{max}$ is the maximum applied stress, and $\sigma_{n}\;$ is the average stress of the structure. The thickness of the coating is denoted by h, and ρ represents the curvature radius of the crack tip. The stress concentration in the substrate material is influenced by the thickness and curvature radius of the coating. Therefore, a thicker coating and a smaller curvature radius result in a higher stress concentration factor, leading to increased stress concentration in the substrate.

In addition to the main crack throughout the coating to the substrate, a crack branch can be seen in Figure 8. The appearance of a crack branch demonstrates that, after the cracking of the hard coating, the propagation of the fatigue crack stopped at the interface of the substrate and coating. A new crack nucleation and crack initiation process began at the interface of the Ti substrate. A small crack just near the main crack, at a very short distance, on the coating surface can also be found. Multiple initiations of fatigue cracks on the surface of the coating are possible due to the brittle and inhomogeneous nature of the coating. This small crack can be seen as a branch of the main crack in the coating.

As shown in Figure 9, there were many cracks in the coating on the same specimen. However, most of them were non-propagating fatigue cracks. These cracks occurred throughout the coating and reached the substrate. The substrates at these crack tips remained undamaged because of the stress relief caused by another crack which had already passed through. It is clear that the fatigue crack at the substrate occurred after the crack in the coating. This phenomenon can also be found in other studies [13,29,30], especially under high cyclic stress [26].

According to the observations above, the fatigue process of this substrate with hard coating can be concluded as follows. When the specimen was under cyclic loading, multiple micro-cracks were initiated near each other in the early stage of the fatigue process (Figure 10a).

They propagated to the interface of the coating and substrate and then stopped (Figure 10b). These multiple coating cracks can also be found in many studies [12,31,32]. As the cyclic loading continued, premature cracking occurred at the substrate, linking to some coating crack tips. The deformation was restrained in a small plastic zone. Among these cracks, the crack with the highest stress intensity factor propagated rapidly and became the dominant fatigue crack (Figure 10c). As a result, the propagation of fatigue cracks in the substrate could be inhibited due to the stress release from other propagated cracks. Most of the cracks stopped growing, and only one capable of continuous propagation eventually led to a fatigue fracture of the specimen (Figure 10d). Decreasing plastic zone size and micro-crack formation were the reasons for the decrease in the crack retardation period, which led to the decrease in fatigue life [12]. 

It is necessary to note that these results and discussion were obtained from the experimental condition provided in this paper. The result may differ if the specimen’s dimensions, materials, processing methods, or load condition are changed. The advantage of this method is that it is easy to set up and better for comparison. However, the disadvantage is that it is difficult to calculate the local stress, so fatigue properties (e.g., S-N curve, fatigue strength) cannot be obtained. More tests with different thicknesses of coatings, loading, coating materials, and intermediate coatings will be conducted in the future, and simulations will be applied using this method.

## 4. Conclusions

In order to investigate the impact of hard coatings on the fatigue properties of metal substrates, a ring-shaped fatigue specimen was utilized. As an example, in this study, specimens of pure titanium with WC-Co coatings were examined. This fatigue specimen was used to successfully compare the fatigue life of coated and uncoated metals.

The simulation results indicate that the presence of a hard coating can alter the stress distribution. Due to the significantly higher Young’s modulus of the coatings, evidently, tensile stress was reduced in the substrate beneath the coating while it was increased within the coating itself. This induced crack initiation at the surface of the coating. A brittle fracture occurred in the coating, which served as the initiation site for fatigue crack formation. 

Based on the conditions investigated in this study, the WC-Co coating has a detrimental effect on the fatigue properties of pure titanium. The presence of coating cracks leads to stress concentration, which usually occurs at the crack tips of the coatings. This accelerates fatigue crack propagation in the substrate material.

After the cracking of the coating, the coating and substrate shared movement from the fatigue machine differently. The coating had no deformation, while the substrate showed continuous deformation. At the interface with ups and downs, a zigzag crack pattern was observed using OM and SEM. This can be attributed to the different motions of the coating and substrate at ups and downs, rendering the substrate more vulnerable. 

After the fracturing of the specimen, multiple micro-cracks could be found at the coating and propagate towards the interface. Stress releases from other crack made them into non-propagating cracks. Ultimately, only one crack propagated throughout the substrate, resulting in a fatigue fracture in the specimen.

## Figures and Tables

**Figure 1 materials-17-00835-f001:**
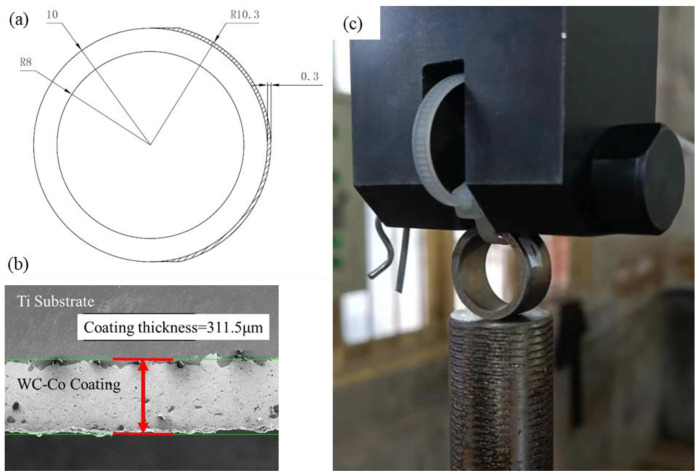
Specially designed ring-shaped fatigue specimen in this work. (**a**) Front view; (**b**) microstructure of coating area; (**c**) setup of the specimen and the fatigue machine in this study.

**Figure 2 materials-17-00835-f002:**
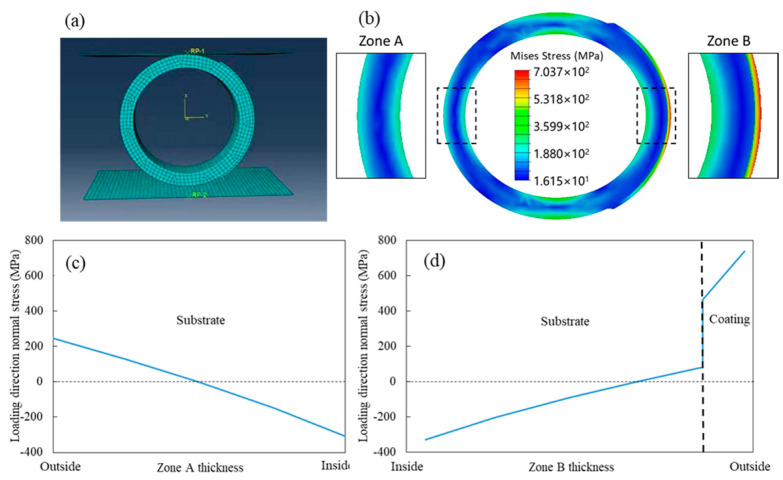
Simulation of the force distribution in the specimen, (**a**) FE model used in this study; (**b**) distribution of Mises stress in fatigue specimen; (**c**) loading direction normal stress of zone A; (**d**) loading direction normal stress of zone B.

**Figure 3 materials-17-00835-f003:**
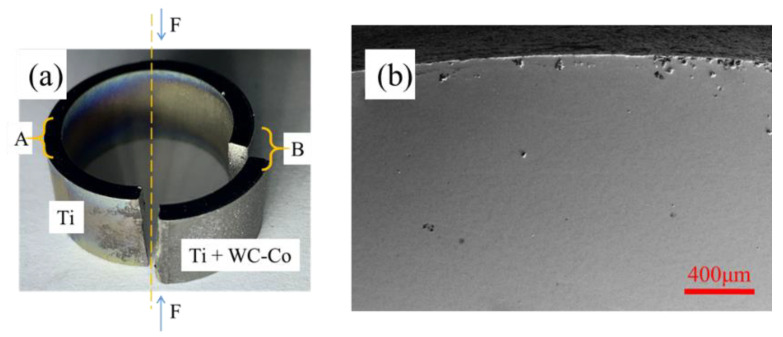
Specimen after fatigue test. (**a**) Overview of the specimen; (**b**) SEM on zone A.

**Figure 4 materials-17-00835-f004:**
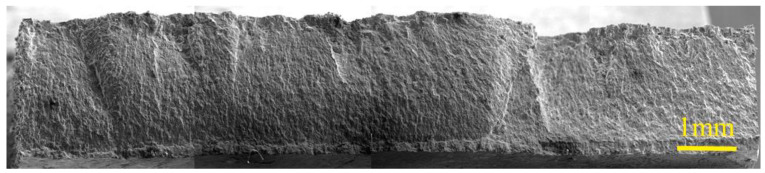
SEM overview on fracture surface of zone B.

**Figure 5 materials-17-00835-f005:**
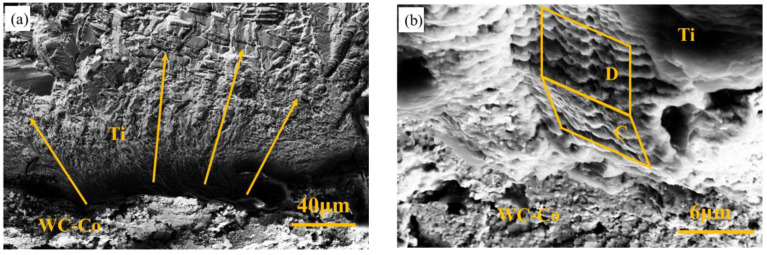
SEM on interface of Ti substrate/WC-Co coating, (**a**) fatigue crack propagation directions; (**b**) fatigue patterns on different facets.

**Figure 6 materials-17-00835-f006:**
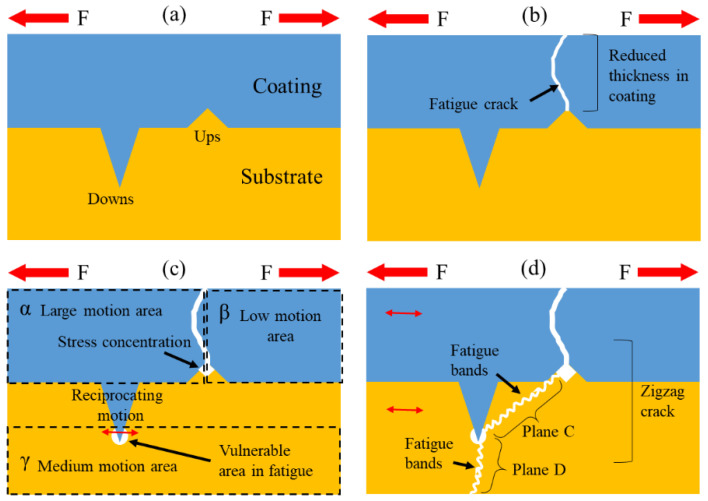
Schematic of mechanism of formation of zigzag fatigue crack propagation at tough interface: (**a**) initial condition, with ups and downs at interface; (**b**) fatigue crack having just crossed through the coating; (**c**) formation of vulnerable area due to different motion conditions; (**d**) formation of zigzag fatigue crack.

**Figure 7 materials-17-00835-f007:**
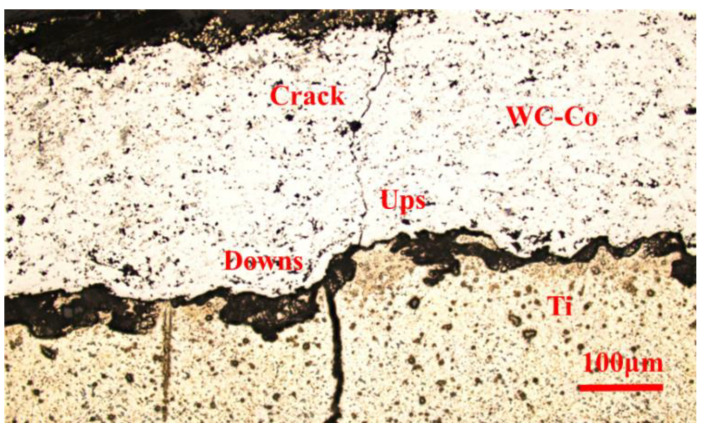
Zigzag fatigue crack observed by optical microscope.

**Figure 8 materials-17-00835-f008:**
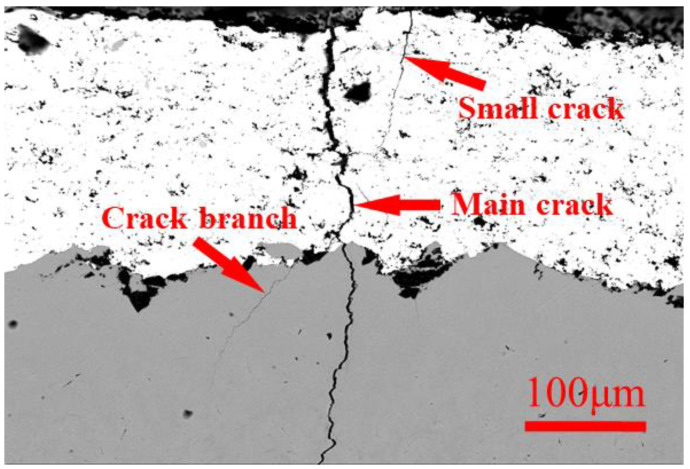
Fatigue cracks throughout the coating and substrate.

**Figure 9 materials-17-00835-f009:**
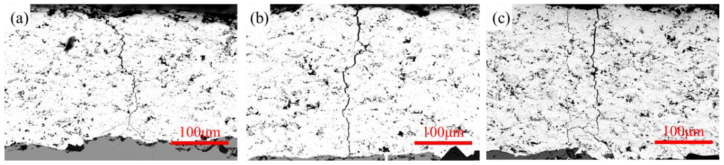
Coating cracks stopped at interface of coating and substrate on the same specimen: (**a**) crack 1; (**b**) crack 2; (**c**) crack 3.

**Figure 10 materials-17-00835-f010:**
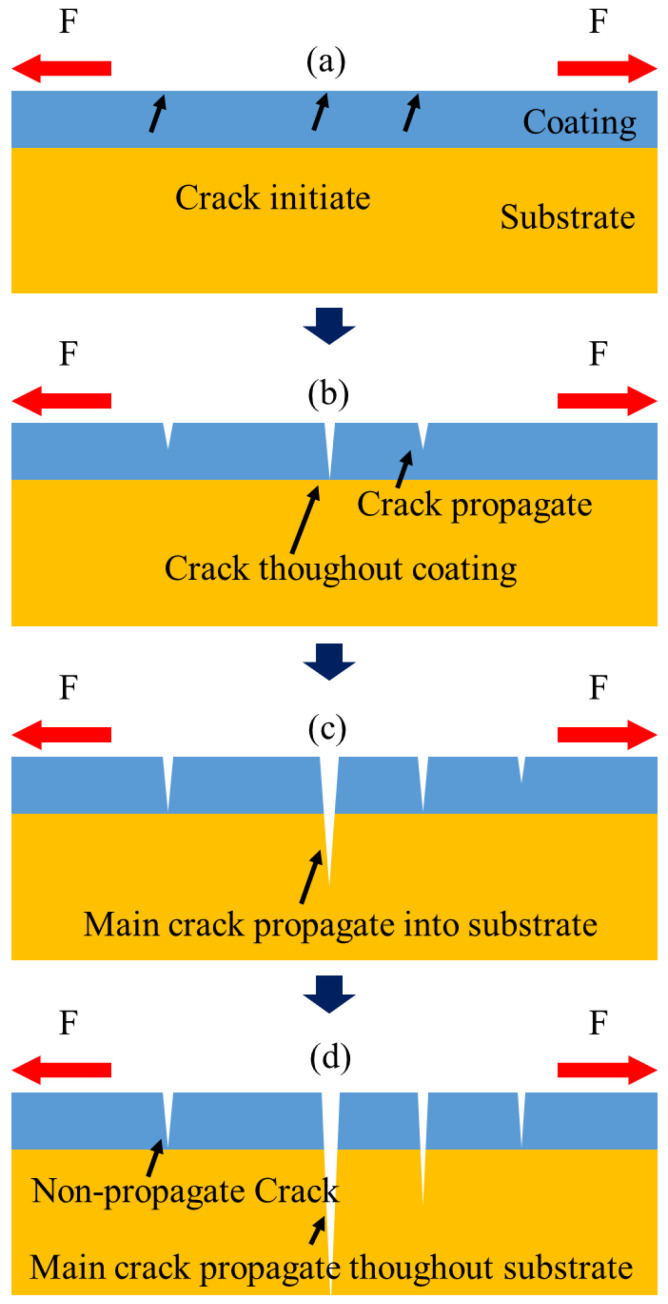
Cross-sectional schematic of fatigue process in coated metals: (**a**) crack initiated at the coating surface; (**b**) fatigue cracks propagated through coating; (**c**) main crack propagated into the substrate; (**d**) main crack stopped other cracks and made them into non-propagating cracks.

**Table 1 materials-17-00835-t001:** Material constants used in FEM modeling.

Materials	Young’s Modulus (GPa)	Poisson’s Ratio
WC-Co	580	0.22
Titanium	108	0.3

## Data Availability

Data available on request from the authors.

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
