# Peer review of "The Influence of Hard Coatings on Fatigue Properties of Pure Titanium by a Novel Testing Method"

_materials, 2024, doi:10.3390/ma17040835_

Round 1

Reviewer 1 Report

Comments and Suggestions for Authors

Concise abstact.

Keywords are missing in the paper - please add them to the text of the manuscript.

Please add a nomenclature to your manuscript - a complete list of symbols, abbreviations and markings that appear in your paper. The nomenclature may even be included at the end of the manuscript. But it is necessary - it has to be at paper.

The introduction is correct, but the way of citing literature needs improvement - it is "[15,16,17,18,19,20,20,22]" and should be "[15-22]". Please check all manuscript in this respect.

On line 99 there is "specimens. [28].” and it should be "specimens [28]." – too many dots at paper. Please check this.

Please include a detailed technical drawing of the specimen used in your manuscript. It is necessary.

The authors refer to the results of a simulation about which we know nothing. Please complete your paper in this regard.

Please make a number of changes and additions to the paragraphs regarding FEM calculations. In the field of numerical modeling, the manuscript requires very extensive development. Please add information to your paper about how many finite elements there were in the model, how many nodes there were, and whether the issue was 3D or 2D. Did the authors use existing axes of symmetry in the modeling? What type of finite elements were used in FEA calculations, how many nodes and numerical integration points were there in one element, what was the size of the finite element? How were critical places modeled in the model? What was the division of the mesh around special places and in the entire model. Please include an enlarged FE model with a visible division of the whole, the applied load and the assumed boundary conditions. Please add an enlargement to the fragment in areas important from the point of view of the paper, so that the finite element mesh can be visually assessed. Please refer to the issue of convergence of the numerical model - how it was tested and how the final FEM model used in the research was established. What material model was used in FEM? What constitutive compound describes it? This is what is missing at manuscript. Was the finite element mesh draft quality or high quality? What material model did the authors use in the calculations? Were the results compared with others of this type for this geometry and material, obtained using e.g. ANSYS, ABAQUS, ADINA?

Please add tables with material constants used in FEM modeling and sample results to your manuscript.

Please elaborate a bit on the conclusions in your paper.

I recommend a major revision.

Comments on the Quality of English Language

 Minor editing of English language required.

Reviewer 2 Report

Comments and Suggestions for Authors

The paper introduces a so-called "Novel Testing Method" to measure the influence of a hard coating on the fatigue properties of materials. The authors focused the paper on pure titanium, and the coating is made of WC-Co. The idea that coating affects the response of a substrate under fatigue is not unknown. I do not understand exactly what the authors saw as a novel. Apart from that, the authors studied fatigue through a simulation, which was not validated by them. This creates an important problem in the experimental design, as the use of simulation without proper experimentation could produce false conclusions. In fact, the experimental procedures are not really performed under any standard, and no testing machine was defined. Even if the authors showed SEM images of a crack propagation process, the lack of a proper standard and a more detailed experimental setup worries me. 

Finally, there is consistency in the observations and the "schematic of mechanisms in the formation of zigzag fatigue crack propagation" that the authors proposed. However, in the reviewer´s opinion, this is mere speculation. More experimental work and deeper analysis should be performed to validate or at least provide more information about this proposed scheme. All these flaws and the need for more experimental work are the reasons to reject the paper.

Comments on the Quality of English Language

The grammar is correct; however, the structure of the paragraphs and the concatenation of the sentences could be improved to help readers understand the main ideas.

Reviewer 3 Report

Comments and Suggestions for Authors

The authors in their work present a study aimed at understanding the effects the impact of hard coatings on the fatigue properties of pure titanium. For the study they proposed a fatigue specimen ring-shaped, coated on one side with hard coating and on the other side without hard coating. The introduction to the subject and the review of the state of the art is pretty good. They explain with detail several aspects of the experimental methodology. From my point of view, the presentation and analysis of the results is profound and therefore the conclusions reached are good. The manuscript has the potential to be published, although the following aspects should be considered prior to publication.

 To authors

1.      It is logical from before the study that if there is a failure in a material coated with a hard coating it may have its origin at the interface of the joint. You used aluminum oxide to improve adhesion of the coating to the substrate. What other surface preparation could be carried out for the purpose of improving adhesion?

2.      Can you explain why you specifically used WC-Co as a coating? what is the function of the Co in the coating? why not just try to coat with WC which is extremely hard?

3.      I consider that the 311-micron thickness of the coating is quite large and hence the origin of the failure due to the fragility of the coating. How do you consider the result would be if you used thinner coatings, even thin films?

4.      How to summarize the effect of hard coating on the fatigue of your study material?

Round 2

Reviewer 1 Report

Comments and Suggestions for Authors

All my comments were included by Authors in new version of the manuscript. I recommend it for publication.

Comments on the Quality of English Language

Minor editing of English language required.

Author Response

Dear reviewer,

Thank you for your thorough review and valuable feedback on our manuscript. We appreciate the time and effort you have dedicated to assessing our work.

For a better quality of this work, a photo of the setup of the fatigue test in this study is added in Figure 1. Language has been improved for better understanding. References have been added to prove our similar findings.

We express our sincere gratitude for your valuable feedback. Thank you for your consideration of our manuscript.

Best regards

Cai Hu

Reviewer 2 Report

Comments and Suggestions for Authors

The paper introduces an already-mentioned "Novel Testing Method" to measure the influence of a hard coating on the fatigue properties of materials.  The authors provided more information about the novelty of the testing method introduced in the paper.  Some critical issues were addressed by the authors too. The section about simulations was also clarified. But still, the authors should conduct more experimental work to provide strong proof that the proposed method operates as it should be. I understand that the work is an ongoing or future plan.

Comments on the Quality of English Language

The paragraphs have been improved, just inor changes should be needed.

Author Response

Dear reviewer,

Thank you for your valuable feedback on our paper. We appreciate your recognition of the clarifications we made regarding the section on simulations. We completely understand your concern regarding the need for more experimental work to validate the proposed method effectively.

It is indeed our intention to conduct further experimental investigations to provide stronger evidence for the functionality and effectiveness of our proposed approach. As you rightly mentioned, this work is part of an ongoing project, and we plan to conduct additional experiments to demonstrate the method's performance in practical applications. More tests with different thicknesses of coating, loading, coating materials, and intermediate coating applied will be conducted in the future and simulation will be applied with this method, which I have added at the end of part Results and discussion. Meanwhile, two references are added to make our finding of multiple cracks more convincing. Several sentences and paragraphs have been modified to improve language.

We genuinely value your suggestion and assurance that we will address this limitation in our future research. We believe that incorporating more experimental results will significantly enhance the credibility and robustness of our findings. Thank you once again for your insightful comment, which will undoubtedly contribute to improving our work.

Best regards

Cai Hu